# Culture Requests and Multi-Drug Resistance among Suspected Urinary Tract Infections in Two Tertiary Hospitals in Freetown, Sierra Leone (2017–21): A Cross-Sectional Study

**DOI:** 10.3390/ijerph19084865

**Published:** 2022-04-16

**Authors:** Julian S. O. Campbell, Saskia van Henten, Zikan Koroma, Ibrahim Franklyn Kamara, Gladys N. Kamara, Hemant Deepak Shewade, Anthony D. Harries

**Affiliations:** 1Ola During Children’s Hospital (ODCH) and Princess Christian Maternity Hospital (PCMH) Laboratory, Ministry of Health and Sanitation, Freetown 00232, Sierra Leone; 2Department of Clinical Sciences, Institute of Tropical Medicine, Nationalestraat 155, 2000 Antwerp, Belgium; svanhenten@itg.be; 3Directorate of Laboratory, Diagnostics and Blood Services, Ministry of Health and Sanitation, Freetown 00232, Sierra Leone; zikankoroma@gmail.com; 4Infection Prevention and Control Unit-Health Security and Emergency Cluster, World Health Organization Country Office, Freetown 00232, Sierra Leone; ibrahimfkamara@outlook.com; 5Joint Medical Unit, Ministry of Defense, Republic of Sierra Leone Armed Forces, Freetown 00232, Sierra Leone; gladysnanillak14@gmail.com; 6Division of Health System Research, ICMR-National Institute of Epidemiology (ICMR-NIE), Chennai 600077, India; hemantjipmer@gmail.com; 7Centre for Operational Research, International Union Against Tuberculosis and Lung Disease (The Union), 2 Rue Jean Lantier, 75001 Paris, France; adharries@theunion.org; 8Department of Clinical Research, Faculty of Infectious and Tropical Diseases, London School of Hygiene and Tropical Medicine, Keppel Street, London WC1E 7HT, UK

**Keywords:** uropathogens, culture requests, antibiotic sensitivity testing, secondary data, children, pregnant women, West Africa, SORT IT, operational research, Sierra Leone

## Abstract

In sub-Saharan Africa, there is limited information about the use of microbiology laboratory services in patients with suspected urinary tract infections (UTIs). This cross-sectional study assessed the requests for urine culture in patients with suspected UTI in two tertiary (maternal and paediatric) hospitals—Freetown and Sierra Leone, during May 2017–May 2021—and determined antimicrobial resistance (AMR) patterns among bacterial isolates. One laboratory served the two hospitals, with its electronic database used to extract information. Overall, there were 980 patients, of whom 168 (17%) had cultures requested and performed. Of these, 75 (45%) were culture positive. During 2017–2019, there were 930 patients, of whom 156 (17%) had cultures performed. During 2020–2021, when services were disrupted by the COVID-19 pandemic, there were 50 patients, of whom 12 (24%) had cultures performed. The four commonest isolates were *Escherichia coli* (36), *Klebsiella pneumoniae* (10), *Staphylococcus aureus* (9), and *Pseudomonas* spp. (6). There were high levels of AMR, especially for trimethoprim-sulfamethoxazole (47%), nalidixic acid (44%), nitrofurantoin (32%) and cefotaxime (36%). Overall, 41 (55%) bacterial isolates showed multidrug resistance, especially *E. coli* (58%), *Pseudomonas* spp. (50%), and *S. aureus* (44%). These findings support the need for better utilization of clinical microbiology services to guide antibiotic stewardship and monitoring of trends in resistance patterns.

## 1. Introduction

Rates of antimicrobial resistance (AMR) are increasing worldwide and the situation has become a global public health threat. As a result of drug resistance, antibiotics lose their effectiveness and become unable to prevent and treat bacterial infections, with potentially devastating consequences [1]. The emergence of antibiotic resistance is linked to irrational, indiscriminate, and incorrect use of antibiotics in human health as well as in the veterinary and agricultural sectors, which is especially a problem in resource-poor countries [2].

Urinary tract infections (UTIs) are amongst the most common infections globally [3]. Most infections are caused by migration of bacteria from the gut, with *Escherichia coli* (*E. coli*) being one of the most common responsible pathogens [4]. Women are at greater risk for UTIs compared to men, due to their anatomical structural differences [4]. Uncomplicated UTIs (restricted to the bladder and urethra) in women may resolve spontaneously. However, if they do not, then antibiotics with limited ecological adverse effects of AMR, such as nitrofurantoin or trimethoprim/sulfamethoxazole (TMP-SMX), are typically used as first line empirical treatment for uncomplicated cystitis [5]. For more serious infections such as acute pyelonephritis, antibiotics such as fluoroquinolones (ciprofloxacin, levofloxacin, ofloxacin) are recommended and are often used on an empirical basis [5].

In line with rising rates of AMR in general, resistance to commonly prescribed antibiotics for UTI is also rising [6]. To optimize treatment of UTIs with antibiotics that are effective, good data on antibiotic resistance patterns that are applicable to the local context are needed because antibiotic resistance patterns of urinary tract pathogens can vary tremendously between regions [7].

While sophisticated surveillance systems exist in some (mainly high-income) countries, in many resource-limited countries, especially in sub-Saharan Africa, surveillance data on antibiotic resistance is scarce. A 6-month study by Kengne M et al. in Chad in 2014 including 160 patients who submitted urine specimens for culture and drug sensitivity found *E. coli* to be the predominant pathogen, with over 60% of all isolates showing multiple drug resistance (defined by by Kengne M et al. as resistance to two or more antibiotics) [8]. A more recent study from Northern Ethiopia on over 1000 urine samples tested between 2012 and 2017 found that nearly 30% were positive for bacterial isolates. *E. coli* was again the most common pathogen in nearly 50% of cases, and multidrug resistance (MDR) was found in almost 90% of isolates [9]. Other published studies from sub-Saharan Africa over the last 5 years have focused on small numbers of patients presenting with suspected UTIs that included students, pregnant women, vulnerable people such as those with diabetes or cancer, and children. Management of UTIs was poor and a call has been made from the West African country Ghana for more utilization of urine culture and antibiotic resistance profiles and more tailored use of antibiotics in response to culture results [10].

At the time this study was planned, there were only two published studies from Sierra Leone on UTIs. The first focused on non-*E-coli Enterobacteriaceae* in outpatients from a small private facility (2013–14) in the district of Bo [11]. In 93 samples, there were 70 *Enterobacteriaceae* urine isolates, of which 86% were MDR and nearly two-thirds produced extended-spectrum B-lactamase (ESBL). The other study (2018) focused on resistance patterns in hospitalized adults with catheter-associated UTIs, while also looking at sputum from patients with hospital-acquired pneumonia [12]. Bacterial isolates were found in 59% of cultured urine samples, with *E. coli* and *Klebsiella pneumoniae* most commonly identified. Overall resistance rates were high (58%) for all ESBL-producing organisms.

More general surveillance studies have not been conducted, so an overview of antibiotic resistance in bacterial isolates causing UTIs in Sierra Leone is currently still lacking. Furthermore, there is no published information about how often urine cultures are performed in patients with suspected UTI. Such information will be vital to support antimicrobial stewardship programs in hospitals in the country, to ensure effective treatments and to limit further spread of antibiotic resistance. Therefore, the aim of this study was to assess the use of urine culture in patients with suspected UTI in two tertiary hospitals in Sierra Leone between May 2017 and May 2021, and to determine antibiotic resistance patterns (including MDR) of the bacterial isolates identified. We also examined the resistance pattern of the antibiotics according to the 2021 World Health Organization (WHO) AWaRe classification [13].

## 2. Material and Methods

### 2.1. Study Design

This was a cross-sectional study using secondary data.

### 2.2. Setting

#### 2.2.1. General Setting

Sierra Leone is situated along the coast of West Africa. It has an estimated population of 7.9 million people in 2019 (based on projections from the 2015 national census), with about 2 million living in Freetown, the capital city [14]. The country is administratively divided into 5 regions and 16 districts. The health system is organized into five tiered levels. The highest levels of care are tertiary referral hospitals, followed by district hospitals that provide secondary levels of care. The third (primary) level of care is provided by peripheral health units (PHUs), which include community health centres (CHCs). Community health posts and maternity and child health posts constitute the fourth and fifth level of care, respectively [15].

#### 2.2.2. Specific Setting

The study was conducted in Ola During Children’s Hospital (ODCH) and the Princess Christian Maternity Hospital (PCMH) in Freetown. Both hospitals are the only tertiary hospitals for referrals of paediatric and maternity cases, respectively. They are situated in Freetown, within the same compound, and are served by the same laboratory that hosts the bacteriology unit.

ODCH is a 270-bed paediatric hospital with an average of 1000 inpatient admissions per month. The clinical staff includes 38 doctors and 208 nurses. PCMH is a 168-bed maternity hospital with an average of 850 inpatient admissions per month. The clinical staff includes 26 doctors and 229 nurses and midwives. The laboratory is staffed by 16 personnel working in four units (Haematology, Microbiology, Biochemistry, and GeneXpert for tuberculosis) and performs analyses on an average of 1870 samples per month.

The bacteriology unit is in charge of antimicrobial sensitivity testing (AST). This unit was established with support from the Centers for Disease Control (CDC), Atlanta, GA, USA, as part of a 2-year project to strengthen post-Ebola laboratory capacity serving the two hospitals. The capability for carrying out bacterial cultures started in May 2017, and it has continued ever since. The project provided the resources to upgrade the unit, train national staff, and provide laboratory equipment and supplies. In addition, a laboratory consultant from the CDC was assigned to the unit to supervise and train staff for 2 years (May 2017–December 2019). The unit was audited in 2019 by the West African Health Organization and obtained one star, with a score of 62%.

#### 2.2.3. Bacteriological Procedures for Suspected Urinary Tract Infections

For patients with a suspected UTI, a urine sample is taken and a request made for microscopy and/or culture. If a request for urine culture is made, a urine sample is placed onto various agar media—such as MacConkey, CLED (cysteine–lactose–electrolyte-deficient), or nutrient agar—and left to incubate at 37 °C for 24 h or longer depending on the result. A sample is diagnosed culture positive based on the growth of bacteria at a high number of colony-forming units (CFUs). For clean-catch urine samples, a sample is considered positive if growth of bacteria is greater than 100,000 CFUs/mL. If no growth is observed after 48 h, the sample is considered as negative. The identification of the bacteria is done using microscopic identification. Biochemical tests are performed to identify the Gram-negative isolates; these include lactose fermentation, urease, indole, and Triple Sugar Iron, while tube catalase and coagulase are used to identify the different species of Gram-positive isolates.

AST is done using the Kirby–Bauer disc diffusion technique, where a standardized inoculum (0.5 McFarland) from a pure culture is seeded onto sterile Mueller Hinton Agar [16]. Excess moisture on the agar surface is absorbed prior to the application of the antimicrobial discs. Gram-positive and Gram-negative antibiotic discs are selected for the bacterial isolates. Plates are incubated aerobically at 35 ± 2 °C for 18–24 h, after which zones of inhibition are measured using callipers. Interpretation is done using the Clinical and Laboratory Standards Institute (CLSI) 2017 guidelines (performance Standards for AST) [16]. Based on the CLSI guidelines, inhibition zones are reported as Sensitive, Intermediate, or Resistant. Antibiotics that are tested are selected based on their common use and the identified bacteriological strains [17,18]. Laboratory methodology included American Type Culture Collection strains and also in-house controls for biochemical and antimicrobial testing. Quality is ensured through a documented standard operating procedure that is reviewed every 2 years, as stated in the quality manual.

#### 2.2.4. Routine Data Quality

Details of all urine samples received at the bacteriology unit of ODCH-PCMH laboratory are entered manually in a report ledger. These are also single-entered daily into a Microsoft Excel database, and updated with the results following confirmatory testing and completion of AST verification. The Laboratory Microsoft Excel database is double-checked by a second laboratory lead, and final verification is made by the consultant medical laboratory scientist. All information is further filtered by the national consultant to verify the results, and while doing so, the consultant conducts a random 10% check on the ledger and requests forms for compliance.

### 2.3. Study Population and Period

Urine samples received in the laboratory from patients at ODCH and PCMH with suspected urinary tract infection between May 2017 and May 2021 formed the study population.

### 2.4. Data Variables, Collection, and Source of Data

Data of patients with suspected UTI were extracted from the urine laboratory database. All information was further filtered by the national consultant to verify the results, who at the same time conducted a random 10% check on the ledger. General variables collected on each patient and the urine samples included the following: age of patient, gender, hospital type, year (2017–2021), urine sample received in the laboratory, urine culture requested, urine culture performed, urine culture positive, bacterial isolates identified, susceptibility to the different antibiotics tested (stratified by resistant and non-resistant), and number of bacteria showing multidrug resistance. Bacterial isolates resistant to at least three antibiotics belonging to different classes of antimicrobial compounds were classified in this study as multidrug resistant [19].

### 2.5. Analysis and Statistics

Data were extracted from the electronic databases, cleaned, and analysed using EpiData analysis software (version 2.2.2.183, EpiData Association, Odense, Denmark). Numbers and proportions were calculated over the total 5-year period, per year and per hospital. Differences between hospitals were compared using the chi-squared test, with levels of significance set at 5% (*p* < 0.05). Bacterial isolates identified along with observed resistance to the tested antibiotics were calculated and are presented as frequencies and proportions. Further analysis was conducted to calculate multidrug resistance and assess resistance in relation to the AWaRe classification [13].

## 3. Results

Of 980 patients, the mean age (standard deviation) was 18 years (12.4), with 41% less than 18 years. There were 21% males and 79% females. The proportions from each hospital were 61% from PCMH and 39% from ODCH.

### 3.1. Urine Culture Requests and Culture Positivity

Of 980 patients with suspected UTI, 168 (17%) had cultures requested, and all of these urine specimens were set up for culture. Of these, 75 (45%) urine specimens were culture positive. The proportion of urine samples for which culture was requested was significantly lower in PCMH when compared to ODCH (9% vs. 30%, *p* < 0.001) (Figure 1).

The annual numbers of patients with suspected UTIs whose urine was received in the laboratory, urine cultures requested/performed, and positive urine cultures are shown in Figure 2. The absolute numbers of samples from patients with suspected UTIs and culture requests were high in 2018 and 2019. The proportions in whom culture was requested varied over the 5 years from 6% (2021) to 33% (2020). In the first 2.5 years (May 2017 to December 2019), there were 930 patients with suspected UTIs, of whom 156 (17%) had requested cultures and also underwent cultures. In the last 1.5 years (January 2020 to May 2021), with services being disrupted by the COVID-19 pandemic, there were 50 patients with suspected UTIs, of whom 12 (24%) had urine cultures requested and performed. Where urine culture was performed, the annual yield of positive bacterial isolates varied each year: 0% in 2017, 49% in 2018, 40% in 2019, and 64% in 2020 (in 2021, only one culture was performed, which was positive).

### 3.2. Bacterial Isolates and Their AMR Patterns

Bacterial isolates and their specific AMR patterns over the whole period of 2017–2021 are shown in Table 1. Altogether, there were 75 bacterial isolates. The four most common isolates identified (in order) were *E. coli* (36), *Klebsiella pneumoniae* (10), *Staphylococcus aureus* (9), and *Pseudomonas* spp. (6), which constituted 81% of all the identified isolates. For those antibiotics commonly used in treating UTIs, overall resistance to the different bacteria was as follows: TMP-SMX, 47%; nalidixic acid, 44%; nitrofurantoin, 32%; cefotaxime, 36%; ciprofloxacin, 15%; and cephalothin 12%. Resistance to both TMP-SMX and nitrofurantoin (commonly recommended for uncomplicated UTI in outpatient settings) was found in 24% of isolates.

Overall, 41 (55%) of the bacterial isolates showed multidrug resistance, with *E. coli* (58%), *Pseudomonas* spp. (50%), and *S. aureus* (44%) showing particularly high rates (Table 1 and Figure 1). Amongst the bacterial isolates combined as “Other” (*Staphylococcus saprophyticus*, *Micrococcus* spp., *Enterococcus faecalis*, and *Acinetobacter* spp.), all showed multidrug resistance. Multidrug resistance ranged from 39% in 2019 to 61% in 2018.

The range of resistance in terms of the WHO AWaRe categories was as follows: Access, 12–47%; Watch, 15–47%; Reserve (Colistin sulphate), 9% (Table 1).

## 4. Discussion

In these two public tertiary hospitals (the paediatric and maternity referral centre) in Freetown, Sierra Leone, the key findings were the low proportion of culture requests received by the laboratory for patients with suspected UTIs, the good yield of bacterial isolates once culture had been set up (with *E. coli*, *K. pneumoniae*, and *S. aureus* as the predominant pathogens), and the high rates of bacterial resistance (including multidrug resistance) to commonly used first-line antibiotics.

The strengths of the study included the long period of observation from 2017 to 2021, and the conduct and reporting of the study according to the STROBE (Strengthening the Reporting of Observational Studies in Epidemiology) guidelines statement [20].

However, there were some limitations. We did not have clinical data (including the number of inpatients and outpatients) nor information on antibiotic treatment or patient clinical outcomes. Identification of some of the bacteria down to species level (for example, for *Pseudomonas* spp. and *Coliform* spp.) was not done due to non-availability of reagents for biochemical tests. For similar reasons, Gram-negative isolates were not assessed for ESBL (for example, ESBL *E. coli*). At various points in time over the 5 years, the lack of discs for commonly used antibiotics in the laboratory prevented a comprehensive AST assessment. In some instances, urine culture was done (based on the presence of white blood cells on urine microscopy) even if a culture request was not made, but the results of these cultures were not systematically collected and are therefore not presented in this study. It would also have been useful to monitor the time taken between receipt of urine samples for culture and the AST results being made available to clinical staff.

Despite the limitations, the study findings have some useful implications for hospital and laboratory practice. First, the low requests for urine culture, which were made in less than a third of patients with suspected UTI in the children’s hospital and less than 10% of patients in the maternity hospital, are a cause for concern. Young children are particularly vulnerable to immediate and long-term complications of UTI, including renal scarring and renal failure, and therefore need prompt effective treatment. A systematic review and meta-analysis, however, shows the high degree of resistance in children to commonly used antibiotics such as ampicillin and trimethoprim [21]. UTI and asymptomatic bacteriuria in pregnancy can also result in considerable maternal and foetal adverse outcomes, such as threatened abortion, preterm labour, and neonatal mortality [22]. As a result, empirical antibiotic treatment—including nitrofurantoin, cephalexin and amoxycillin—is commonly used. A recent systematic review and meta-analysis of UTIs in pregnant women, however, found a pooled global prevalence of ESBL-*Enterobacteriaceae* of 25%, with the highest rates in sub-Saharan Africa [23]. Failure to identify urinary tract AMR in these vulnerable women may lead to failed treatment. The study was not designed to explore why urine cultures were not requested, but possible reasons included a perceived lack of confidence in the laboratory always being able to conduct a comprehensive AST and concerns about length of time from culture requests to results. The number of urine culture requests received in the laboratory also fell drastically during the COVID-19 pandemic, wherein the whole hospital complex was shut down for a significant period. This was due to various factors such as illness and self-isolation occurring amongst staff and patients.

The important implication of this finding of low requests for urine culture is the need to strengthen linkages between clinical departments and the laboratory as well as strengthening laboratory performance itself. Clinicians need to be assured that if they request urine culture, they will receive results quickly so as to inform the most effective treatment. Regular laboratory–clinician interactions and other confidence building strategies may be considered. It was reassuring that 100% of culture requests in the two hospital were honoured, although we have no information about time taken to receipt of results. The logistic problems in the laboratory with shortages of biochemical tests and antibiotic discs are not unique to Sierra Leone, but focused attention is needed to ensure that there are uninterrupted supplies. Two years of support from US-CDC in the post-Ebola period may not have been enough, and lengthier assistance (for 5 years or more) might have been better.

Second, in line with other studies from Sierra Leone, West Africa, and other parts of Africa, *E. coli* and *K. pneumonia* were the most common uropathogens identified [12,24,25,26,27]. We were surprised at the isolation of *S. aureus* in nine specimens. However, although this bacterium is perceived to be uncommon in urine specimens, it was found in 31% of patients with UTI in Uganda [28], 10% of patients in Ethiopia [27], and in 26% of children with UTI in Cameroon [29].

Third, just over half of the uropathogens detected showed multidrug resistance, which was particularly apparent with *E. coli*. The rate of multidrug resistance among all the bacterial isolates was lower than that reported among non-E. *coli Enterobacteriaceae* in a small private facility in Sierra Leone [11], although similar to the 47% rate reported from a teaching hospital in Nigeria [24]. The prevalence of resistance to commonly used antibiotics such as ciprofloxacin (15%), nitrofurantoin (32%), and TMP-SMX (47%), although of concern, was lower than previously reported from the same two hospitals when they analysed AMR patterns from various different samples such as stool, urine, cerebrospinal fluid, pus wounds, pleural fluid, and high vaginal swabs [30]. The previous reports from Sierra Leone in other hospitals have shown high levels of resistance to TMP-SMX among uropathogens [12], a high prevalence of ESBL-producing uropathogens [11], and high levels of multidrug resistance [11], and our study adds to the evidence base. Sierra Leone is not alone in facing these problems, as high levels of resistance to TMP-SMX among uropathogens have also been reported from Ethiopia (67–86%), Egypt (58%), and Nigeria (89–100%) [26,27,31,32].

The high prevalence of multidrug resistance and resistance to commonly used antibiotics further adds to the argument for better utilization of clinical microbiology services for antimicrobial stewardship. This includes implementing the Global AMR use surveillance system (GLASS) [33]. Once implemented, the data from GLASS can be used for modifying standard treatment guidelines for empirical treatment of suspected UTI while awaiting culture AST results. Experience from countries such as Ghana and Thailand emphasize the use of clinical microbiology for setting up and improving antimicrobial stewardship [10,34]. Once an antimicrobial stewardship programme is set up, it needs to be fully supported and regularly evaluated.

Finally, a recent systematic review estimated nearly five million deaths globally associated with bacterial AMR and 1.3 million deaths attributable to bacterial AMR in 2019 [35]. The highest all-age death rate attributable to AMR was in West Africa. The review highlighted the serious data gaps in many low-income settings, emphasizing the need to expand laboratory capacity and data collection systems to better understand this important and growing threat to human health. This work from the two referral hospitals in Sierra Leone will contribute to the national and global evidence base. However, much more needs to be done to strengthen laboratory infrastructure and create resilient health systems that can survive despite challenges such as the COVID-19 pandemic. The Lancet Editorial calling on the Global Fund to embrace AMR as one of its core responsibilities is timely and welcome, and at the country level in Sierra Leone, this would help to improve antibiotic stewardship [36].

## 5. Conclusions

In two public referral hospitals in the capital of Sierra Leone, there were low numbers of culture requests in urine samples received in the laboratory from patients with suspected UTI. Urine culture requests were always honoured and there was a generally high yield of positive cultures, with *E. coli*, *K. pneumoniae*, and *S. aureus* as the predominant pathogens. Levels of AMR were high, especially to commonly used antibiotics, and over half of bacterial isolates showed multidrug resistance. The laboratory requires further strengthening to build confidence among the clinicians, which has been discussed. The setting up of GLASS and antimicrobial stewardship programmes is urgently required to control and monitor trends of AMR.

## Figures and Tables

**Figure 1 ijerph-19-04865-f001:**
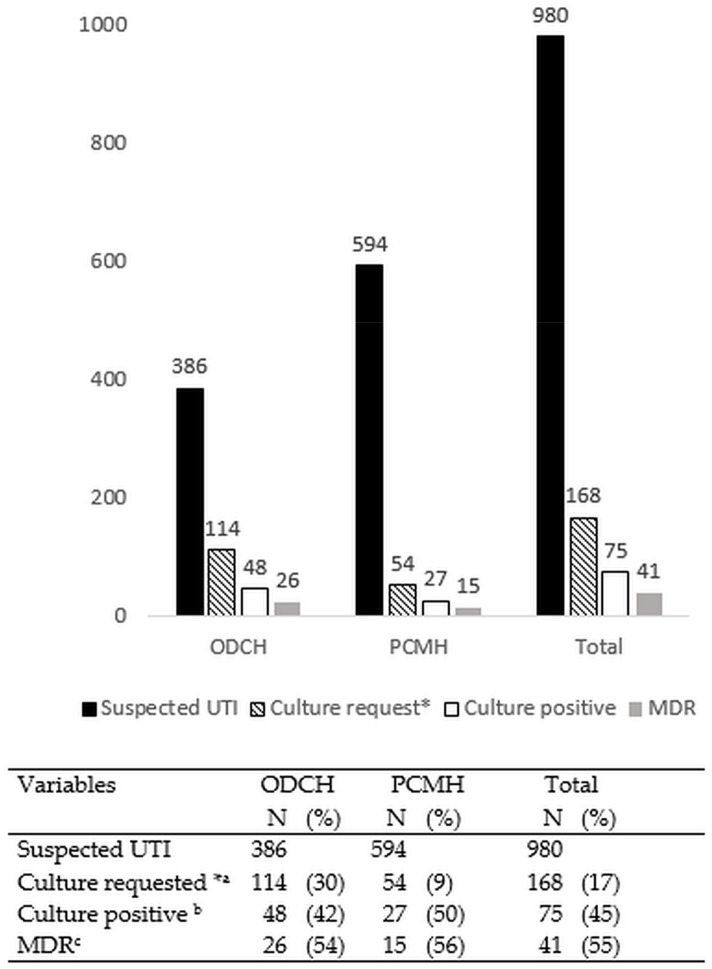
Numbers of patients with suspected urinary tract infections, culture requests, and those processed through to positive cultures in the two hospitals in Freetown, Sierra Leone, between May 2017 and May 2021. UTI—urinary tract infection; MDR—multidrug resistant; * all culture requests underwent urine culture; ^a^ denominator is suspected UTI; ^b^ denominator is culture requested; ^c^ denominator is culture positive.

**Figure 2 ijerph-19-04865-f002:**
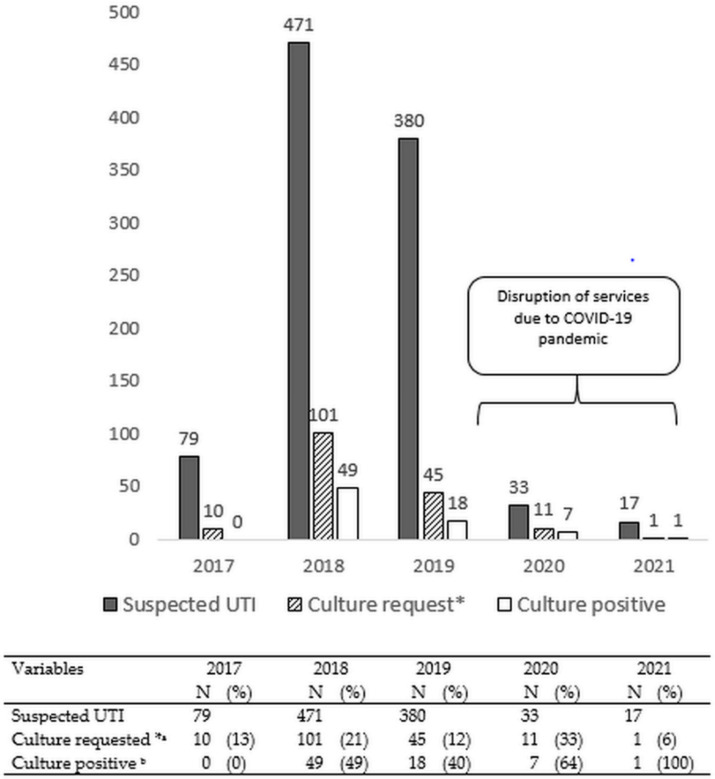
Numbers of patients with suspected urinary tract infections, culture requests, and those processed through to positive cultures in the two hospitals in Freetown, Sierra Leone, between May 2017 and May 2021. UTI—urinary tract infection; * all culture requests underwent urine culture; ^a^ denominator is suspected UTI; ^b^ denominator is culture requested.

**Table 1 ijerph-19-04865-t001:** Bacterial isolates and their antimicrobial resistance patterns in culture-positive urine samples in two hospitals in Freetown, Sierra Leone, between May 2017 and May 2021 [N = 75].

Isolates and Their Resistance Patterns	*E. coli*		*K. pneumoniae*		*S. aureus*		*Pseudomonas* spp.		Coliforms		*Proteus* *mirabilis*		Other *		Total	
	n	(%)	n	(%)	n	(%)	n	(%)	n	(%)	n	(%)	n	(%)	n	(%)
Total isolates	36		10		9		6		4		3		7		75	
Resistant to																
Cephalothin ^A^	5	(14)	-	-	-	-	-	-	1	(25)	-	-	3	(43)	9	(12)
Gentamicin ^A^	3	(8)	1	(10)	2	(22)	-	-	-	-	2	(67)	3	(43)	11	(15)
Ciprofloxacin ^Wa^	6	(17)	1	(10)	-	-	-	-	-	-	1	(33)	3	(43)	11	(15)
Colistin Sulphate ^Re^	2	(6)	-	-	2	(22)	-	-	1	(25)	-	-	2	(29)	7	(9)
Nalidixic acid	15	(42)	2	(20)	4	(44)	1	(17)	1	(25)	3	(100)	7	(100)	33	(44)
Nitrofurantoin ^A^	10	(28)	4	(40)	-	-	4	(67)	2	(50)	1	(33)	3	(43)	24	(32)
TMP-SMX ^A^	17	(47)	4	(40)	-	-	3	(50)	4	(100)	2	(67)	5	(71)	35	(47)
Cefotaxime ^Wa^	14	(39)	2	(20)	6	(67)	2	(33)	-	-	-	-	3	(43)	27	(36)
Oxacillin ^A^	-	-	-	-	-	-	-	-	-	-	-	-	1	(14)	1	(1)
Erythromycin ^Wa^	-	-	-	-	-	-	-	-	-	-	-	-	1	(14)	1	(1)
Imipenem ^Wa^	3	(8)	-	-	-	-	2	(33)	-	-	-	-	1	(14)	6	(8)
Multi-drug resistance	21	(58)	2	(20)	4	(44)	3	(50)	1	(25)	3	(100)	7	(100)	41	(55)

Column percentages: *E. coli*—*Escherichia coli*; *S. aureus*—*Staphylococcus aureus;* TMP-SMX—trimethoprim-sulfamethoxazole. ^A^—Aware group, ^Wa^—watch group, ^Re^—reserve group of antibiotics [13]. -: the antibiotic disks were not used on the bacterial isolate. * Other: *Staphylococcus saprophyticus* (2), *Micrococcus* spp. (2), *Enterococcus faecalis* (2), *Acinetobacter* spp. (1).

## Data Availability

The dataset used in this paper has been deposited at https://doi.org/10.6084/m9.figshare.19096244 (accessed on 31 January 2022) and is available under a CC BY 4.0 license.

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
