# Peer review of "Culture Requests and Multi-Drug Resistance among Suspected Urinary Tract Infections in Two Tertiary Hospitals in Freetown, Sierra Leone (2017–21): A Cross-Sectional Study"

_ijerph, 2022, doi:10.3390/ijerph19084865_

Round 1

Reviewer 1 Report

Minor revisions:

Abstract:

Line 28: add a space after 168

Lines 29/30/31/35: space after the numbers prior to the parenthesis

Line 35: delete of

The last sentence may make more sense with “trends in resistance patterns.”

Introduction:

The background and justifications for the study were easy to follow and understand.

Materials and methods:

2.2.1: The health system is described as five tiered levels, but only three are explained. Why were the other levels not noted?

Line 179: “Bacterial isolates resistant to at least three antibiotics belonging to different classes of antimicrobial compounds were classified in this study as multidrug-resistant [19]” contradicts line 69 “showing multiple drug resistance (defined in this study as resistance to two or more antibiotics)”

Line 189: missing a period

Discussion

If 100% of requested tests were honored in the laboratory, then the clinician’s concern that a request may not be fulfilled does not seem like a fair justification for such low culture request percentages. Are there any other possible reasons that so few culture requests were made?

Conclusions

How will the laboratory-clinician trust be increased moving forwards?

Author Response

#REVIEWER ONE

Comments

Line 28: add a space after 168

Lines 29/30/31/35: space after the numbers prior to the parenthesis

Line 35: delete of

The last sentence may make more sense with “trends in resistance patterns.”

Response:

Thank you for your comments

We have incorporated these changes the revised manuscript.

Comment 

Introduction:

The background and justifications for the study were easy to follow and understand.

Response

Thank you

Comment

Materials and methods:

2.2.1: The health system is described as five tiered levels, but only three are explained. Why were the other levels not noted?

Response:

Thank you for your comments. The five tier health system is as follows

  1. MCHP
  2. CHP- Community Health Post
  3. CHC- Community Health Centre
  4. District (secondary) Hospital
  5. Tertiary Hospital

We have updated this information in the general settings (revised manuscript)

Comment:

Line 179: “Bacterial isolates resistant to at least three antibiotics belonging to different classes of antimicrobial compounds were classified in this study as multidrug-resistant [19]” contradicts line 69 “showing multiple drug resistance (defined in this study as resistance to two or more antibiotics)”

Response:

Thank you for your comments

Line 69 has a citation at the end (ref 8 by Kengne M et al). This study gave the extent of multiple drug resistance, here they used 2 or more antibiotics as the definition, hence, we clarified it in parenthesis. However the standard definition of multidrug-resistant is resistant to at least three antibiotics belonging to different classes of antimicrobial compounds (which we have used). For clarity, we have revised the sentence as follows

“A 6-month study by Kengne M et al in Chad in 2014 among 160 patients who submitted urine specimens for culture and drug sensitivity found E.coli to be the predominant pathogen with over 60% of all isolates showing multiple drug resistance (defined by Kengne M et al as resistance to two or more antibiotics) [8].”

Comment:

Line 189: missing a period

Response:

Thank you for your comments

We have incorporated the edit suggested.

Comment

Discussion

If 100% of requested tests were honored in the laboratory, then the clinician’s concern that a request may not be fulfilled does not seem like a fair justification for such low culture request percentages. Are there any other possible reasons that so few culture requests were made?

Response:

Thank you for your comments

That the use of antibiotic susceptibility testing in these two facilities started in 2017 (with the support of CDC). Before this antibiotic susceptibility testing was not available in these two facilities. So, we speculate that it takes time for the clinicians to change their leant behavior over a period of time. In addition, the on certain instances, some antibiotic discs were not available. So the antibiotic susceptibility report did not have the sensitivity pattern for some of the commonly used antibiotics. These two could be the reason for low culture requests. We have discussed this in 297-307 of revised manuscript with track changes.

Comment

Conclusions

How will the laboratory-clinician trust be increased moving forwards?

Response:

Thank you for your comment

This can be done through a laboratory clinician interface and other confidence building strategies as mentioned in lines 297-307 of revised manuscript with track changes.

Reviewer 2 Report

Dear Authors, 

Thank you for submitting your work to this journal.

In an infrastructure limited setting it is appreciated that this study highlights the gaps in diagnosing and treating common infections. 

Please find below suggestions/comments: 

1] Methodology: 

  1. Please explain in brief if the laboratory methodology included quality control strains for biochemicals and antimicrobial testing.
  2. How are the laboratory methods quality assured?
  3. Only one culture plate has been used for each specimen- why?               Both of these media do not support the growth of ALL uropathogens- so please address this limitation. 
  4. Some of the drugs that have been included in Table 1, may need to be excluded as these are not agents used in UTI- Chloramphenicol, Tetracyclines and Kanamycin-as its reserved for other infections.

Statistics: 

  1. Population that these hospitals cater to in comparison to the entire population in Freetown. Please clarify if they serve the entire 2 million population as mentioned, and further if this represents 1/4th the population of Sierra Leone.
  2. Based on the title- this is a cross-sectional study. - The only difference highlighted is in the age of the patients (as it is presumed) and the number of suspected and confirmed UTI's. There is no difference highlighted in terms of the organism spectrum and the antimicrobial susceptibility. Please do explain this difference or include the information in Table 2. 
  3. What was the inclusion/exclusion criteria- did it include all lab requests/clinically suspected cases irrespective of a patients number of visits/specimens submitted?- please clarify.

Thank you.

Author Response

#REVIEWER TWO

Comment

Please explain in brief if the laboratory methodology included quality control strains for biochemicals and antimicrobial testing.

Response:

Thank you for your comments

American Type Culture Collection (ATCC) strains and also in-house controls are available.

Please see 159-162 lines of revised manuscript with track changes.

Comments:

How are the laboratory methods quality assured?

Response:

Thank you for your comments.

All methods are followed by a documented standard operating procedure (SOPs) that are reviewed every two years as stated in the quality manual. Please see lines 159-162 of revised manuscript with track changes.

Comments:

Only one culture plate has been used for each specimen- why?  Both of these media do not support the growth of ALL uropathogens- so please address this limitation. 

Response:

Thank you for your comments

We also used nutrient agar and this has been added in line 141 of revised manuscript with track changes.

Comment

Some of the drugs that have been included in Table 1, may need to be excluded as these are not agents used in UTI- Chloramphenicol, Tetracyclines and Kanamycin-as its reserved for other infections.

Response:

Thank you for your comments. We agree and we have deleted these from Table 1.

Comment:

Statistics: Population that these hospitals cater to in comparison to the entire population in Freetown. Please clarify if they serve the entire 2 million population as mentioned, and further if this represents 1/4th the population of Sierra Leone.

Response:

Thank you for your comments

Two million is the total population of Free Town and this is 1/4th of the population of Sierra Leone. These two hospitals provide services (cater to) the entire population of Free Town  and other parts of the country. We have confirmed the same.

Comments:

Based on the title- this is a cross-sectional study. - The only difference highlighted is in the age of the patients (as it is presumed) and the number of suspected and confirmed UTI's. There is no difference highlighted in terms of the organism spectrum and the antimicrobial susceptibility. Please do explain this difference or include the information in Table 2.

Response:

Thank you for your comments. We did consider including information in terms of differences in the organism spectrum and the antimicrobial susceptibility (by the age of patients), but the numbers of bacterial isolate for each organism for too low to make any meaningful comparisons. We hope this is fine.

Comments

What was the inclusion/exclusion criteria- did it include all lab requests/clinically suspected cases irrespective of a patients number of visits/specimens submitted?- please clarify.

Response:

Thank you for your comments

We do not have a single moment when we received two samples from single patient. Urine samples received in the laboratory from patients at ODCH and PCMH with suspected urinary tract infection between May 2017 and May 2021 formed the study population.